# Adjusting the family's life: A grounded theory of caring for children with special healthcare needs in rural areas, Thailand

**Katemanee Moonpanane**[1]*, **Salisa Kodyee**[1], **Chomnard Potjanamart**[1], **Eva Purkey**[2]

**1** School of Nursing, Mae Fah Luang University, Chiang Rai, Thailand, **2** Department of Family Medicine, Faculty of Health Science, Queen's University, Kingston, Canada

* katemanee.moo@mfu.ac.th

## Abstract

This study aims to understand the experiences of families of children with special healthcare needs in rural areas in Thailand. Grounded theory (GT) was employed to understand families' experiences when caring for children with special healthcare needs (CSHCN) in rural areas. Forty-three family members from thirty-four families with CSHCN participated in in-depth interviews. Interviews were recorded and transcribed. The constant comparative method was used for data analysis and coding analysis. Adjusting family's life was the emergent theory which included experiencing negative effects, managing in home environment, integrating care into a community health system, and maintaining family normalization. This study describes the process that families undergo in trying to care for CSHCN while managing their lives to maintain a sense of normalcy. This theory provides some intervention opportunities for health care professionals when dealing with the complexities in their homes, communities and other ambulatory settings throughout the disease trajectory, and also indicates the importance of taking into consideration the family's cultural background.

## Introduction

Advanced medical technology has increased the survival rate of children from traditional killers such as infectious disease and has improved the survival of a greater number of children with special health care needs (CSHCN) [1, 2] who require more complex care for their physical, developmental, behavioral, or emotional differences than their typically developing peers [3–5]. Special healthcare needs can include those related to physical, intellectual, and developmental disabilities as well as long-standing medical conditions, such as heart disease, asthma, diabetes, thalassemia, and end-stage renal disease [ESRD] and may involve medical equipment dependencies [6, 7]. These conditions affect many aspects of the children's lives resulting in frequent emergency visits, prolonged hospitalizations and high rates of readmission [8–10].

The growing number of CSHCN is a global concern [11, 12]. In Thailand, there are approximately 3.9–11.6/1000 live births annually in which the baby is recognized as having serious health conditions [13–15]. These children consume more healthcare resources, incur higher healthcare costs, have the potential to face more health complications and have a considerably higher mortality rate than other populations [14, 16].

University Health Sciences and Affiliated Teaching Hospitals Research Ethics Board (email: traq@queensu.ac; Tel +1 613-533-6000 ext. 74814) and Mae Fah Luang University Ethics Committee (email: rec.human@mfu.ac.th; Tel +66 53 91 7171) for researchers who meet the criteria for access to confidential data.

**Funding:** This work was supported by a grant from The Canadian Queen Elizabeth II Diamond Jubilee Scholarships (QES), Canada. The funders had no role in study design, data collection and analysis, decision to publish, or preparation of the manuscript.

**Competing interests:** The authors have declared that no competing interests exist.

**Abbreviations:** CP, Cerebral palsy; CSHCN, Children with special health care needs; DS, Down syndrome; ESRD, End-stage renal disease; GT, Grounded theory; ICF, The International Classification of Functioning, Disability and Health; SDHPH, Sub-district health promotion hospital; UC, Universal coverage; VHV, Village health volunteers.

In the Thai context, parenting roles attributed to fathers, mothers, and grandparents are interdependent and [17–19] are involved in a caring process for CSHCN that seeks to manage the diseases or equipment as well as to maintain family roles and relationships. However, little is known regarding the exact burden and processes of caring for CSHCN in Thailand, and there are no special supportive programs that directly aim to reduce the burden and adverse effects on the families of CSHCN [19–21].

The proposed study is the first qualitative study employing GT [22] to explore the experiences of the families of CSHCN in rural Thailand. This study aims to understand the perspective of families' caregiver and to capture families' experiences when caring for CSHCN at home, in a hospital, or in a community setting. In addition, this study seeks to identify gaps in care, opportunities, challenges, values, and supportive factors to improve quality of life, quality of care, and accessibility. Additionally, the findings are expected to contribute to guidance, suggestions, and substantive theory for future empirical clinical and community interventions.

## Methods

We used GT developed by Strauss and Corbin [22] for this study. GT is a qualitative approach that focuses on understanding the complexity of a phenomenon from the perspectives and experiences of participants. It is rooted in symbolic interaction and the constructivist paradigm, which proposes to identify categories and to develop a theoretical model to link their relationships and interactions with others. A better understanding of how families experience and respond when caring for CSHCN is important for developing strategies to overcome the problems, stigma, and life circumstances which they may experience, and to maintain family function.

### Setting

Chiang Rai province is located in the northernmost part of Thailand and shares a border with Myanmar. Participants were recruited from four disadvantaged districts (Mae Suai, Mae Fah Luang, Mae Sai and Thoeng) and were approximately 90 km or 2 hours away from the provincial capital. A recent publication showed that residents of these areas have significantly lower formal education and incomes compared to residents of urban areas due to the geographical isolation, poverty, and lack of education.

### Study population

The study participants were family members (fathers, mothers, grandparents, and siblings) living with CSHCN in a rural area in Chiang Rai, Thailand with cultural diversity and mix of ethnic minority group such as Thai, Akha, and Lahu. The researcher recruited the participants who met the inclusion criteria by sending the invitation to participate to the Sub-District Health Promotion Hospital (SDHPH). Fitty-two family eligible for the study and were invited to participate during their previously scheduled appointments. Participants were selected based on the following criteria: 1) participants were primary caregivers for those children who present with typical physical, developmental, behavioral, or emotional disability e.g. Down syndrome, autism spectrum disorder (ASD), cerebral palsy, thalassemia, epilepsy, systemic lupus erythematosus (SLE), end-stage renal disease (ESRD); 2) participants were at least 18 years old; and 3) participants are in stable condition both physically and mentally. Potential fitty two family were given a written information regarding the nature of the study including the purpose and the fact that they have the right to withdraw from the study at any point during the study. If the participant qualified and willing to participate, they were asked to sign on the consent form.

## Ethical approval

All research protocols were conducted in accordance with relevant guidelines and regulations. All instruments and methods were approved by the Queen's University Health Sciences and Affiliated Teaching Hospitals Research Ethics Board (no.6024321) and Mae Fah Luang University Ethics Committee (REH-61201). All participants who met the inclusion criteria were informed of the purpose, risks, confidentiality, and benefits before they voluntarily decided to participate in this study. Informed written consent was obtained from all participants.

## Data collection

Data were collected through in-depth interviews with family members by using a semi-structured interview guide to elicit data related to the experience of caring for CSHCN in the context of living in a rural area in Thailand. Interviews took place between April and August 2019 in Thai dialect lasting between 65 and 110 minutes. Interviews took place in locations preferred by each family, including SDHPHs, homes and community centers. Most of the participant were interview once during that time and had the choice of having the interview conducted together with other members of their family, or on their own. For two participants, more than one interview was carried out in order to further explore issues emerging from the data.

Questions were open-ended and included questions such as "Can you talk about your experiences of raising a CSHCN when he/she was young"; "Can you share with me how you dealt with the situation when your child was given the diagnosis?"; "What kind of support have you had over the years with your child?". As participants narrated their stories, probing questions were asked to assist the participants to explain the situation and all interviews were audio-recorded and subsequently transcribed with the participants' permission.

A co-researcher also conducted participant observations to collect data and generate theoretical accuracy rooted in the social reality of participants' everyday lives. Participant observations included observing CSHCNs receiving care by study participants and other family members, or receiving information from health care professionals. This confirmed an understanding of the essential experience and process of living with CSHCN in rural areas in Thailand. Data were comprised of verbatim transcriptions of 45 in-depth-interviews and observation notes.

## Data analysis

Data analysis began by reviewing the verbatim transcripts to ensure accuracy. The analysis process followed the guidelines of GT put forward by Strauss and Corbin [22], which consist of three steps: open coding, axial coding, and selective coding assisted by the NVivo 12.0, which allowed the systematic treatment of the data, and the ability to keep explicit track of all coding steps. Line-by-line coding helps to identify the initial meaning, concepts and dimensions of the families' experiences. For example, when family members explained the feelings related to having a child with a profound disease and needing specific treatments, they identified several burdens and tasks related to the child's conditions. Later in the interviews, we asked more specific questions about living with a CSHCN in a rural area, and gave participants the opportunity to describe their perspectives in narrative. As new dimensions were identified, the earlier data were re-analyzed to determine if what was found in later interviews existed in the earlier interview data. Through constant data comparison, conceptually clear categories were established that were well represented in information obtained.

## Rigor

Rigor of the study was enhanced by several strategies: establishing credibility, prolonged engagement with participants, participant observation, member checking, establishing an audit trail, and peer debriefing [23]. The authors served as peer debriefers for each other by checking each other's bias as a researcher and corroborating analysis of data. Since the authors have different academic backgrounds, reading transcriptions and observation notes and discussing each other's interpretations of the data helped to identify implicit biases towards the data. This also helped to avoid arbitrary implementation of interview procedures; in particular, reviewing how the interviewer implemented the interview questions and subsequent probes and exploring whether there was any influence of the interviewer on the participants' responses were particularly helpful. Five participants reviewed the summary of study results to conduct a member check. These participants confirmed that the summary results captured the essence of their experiences. Researchers worked intensively on data analysis to confirm the findings, and materials related to the study were comprehensively collected for an audit trail. Finally, contextual information about the research findings was described in as much detail as possible so that readers could assess whether or not the findings were transferable.

## Results

Fitty-two family eligible for the study and forty-three family members from 34 families (15 mothers, 6 fathers, 12 grandmothers, 8 sisters, and 2 brothers) consented to participate in the study. In total 6 families included interviews with both the mothers and fathers; 7 families mothers and grandmothers; 8 families grandmothers and sisters; and in 13 families interviews were completed by only one member (mother, brother, grandmother or sister). Participants characteristic are provided in Table 1.

**Table 1.**

| Characteristic | n (= 20) | % |
|---|---|---|
| **Gender** | | |
| • Male (Mean age 39.88, S.D. = 21.03) | 8 | 18.60 |
| • Female (Mean age 15.32, S.D. 15.32) | 35 | 81.40 |
| **Marital status** | | |
| • Single | 9 | 20.93 |
| • Married | 28 | 65.12 |
| • Separated | 4 | 9.30 |
| • Widowed | 2 | 4.65 |
| **Education** | | |
| • No formal education | 12 | 27.91 |
| • Elementary school | 19 | 44.19 |
| • High school or vocational school | 12 | 27.19 |
| **Ethnicity** | | |
| • Akha | 31 | 72.09 |
| • Lahu | 5 | 11.63 |
| • Thai | 7 | 16.28 |
| **Number of years in caring for the CSHCN** | | |
| • < 3 years | 8 | 18.60 |
| • 3–5 years | 5 | 11.63 |
| • 6–10 years | 12 | 27.91 |
| • > 10 years | 18 | 41.86 |

### Emergent theory: Adjusting the family's life

Family overall experience of caring for CSHCN can be described as "Adjusting the family's life". This term originated from a family members' narrative when caring for a CSHCN in a rural area. Adjusting the family's life (Fig 1) comprised four themes and eight sub-themes which are illustrated in a model. These themes and subthemes illustrate the experiences and processes of families who care for CSHCN in rural areas in Thailand, and comprises four major processes.

**1. Experiencing life's circumstances.** The families report negative effects to their life when care for CSHCN. These impacts are consistent with many that are described in the literature. Most participants reported impacts on their physical and psychological well-being as well as on their relationships. Some families were separated due to the untreatable disease and stigmatization that led them developing negative emotions, withdrawing from social situations, and offering limited opportunities for themselves to positively interact with others.

*Disrupting family life and routine.* Having to raise a CHSCH is an unexpected event and evoked difficult emotions. Family members reported feeling overwhelmed, chaotic, and uncertain. They tried to support and encourage other family members who were caring for CSHCN however, it was difficult for those families who live in rural areas where supports are lacking.

*"The doctor said we should visit the hospital regularly because of the disease but it hard for us. When we have to visit the doctor I have to stop working, . . .my older child left the school . . ..it is too sad to see this boy working in the field and taking care for little boy instead of studying in the school. . . don't know if it is good for all of us? But we survived."*

## Adjusting the family's life

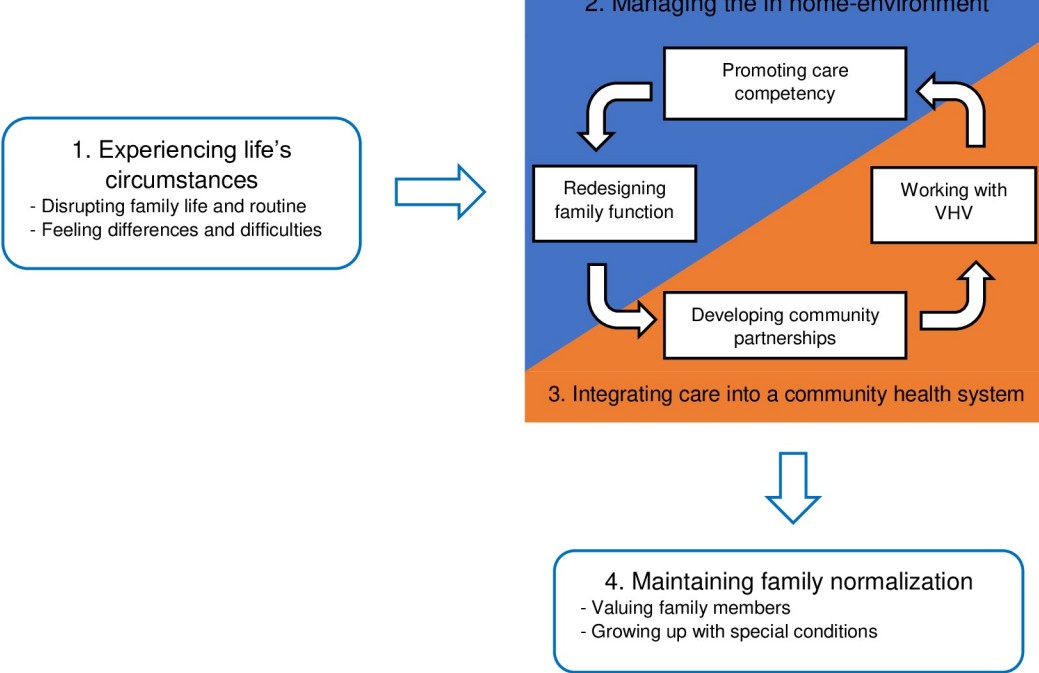

**Fig 1. Adjusting the family's life.**

*Feeling differences and difficulties.* For this study, participants were more likely to describe the experience of living with CSHCN as difficult, hard, tough, and painful situations. However, religions led them to recognize the nature of life, especially Buddhism, realized that with the arising of birth, there is the arising of ageing, sickness, and death. Therefore, family members endeavored to create "good karma" by making strong statements about how they had to learn to overcome these difficulties. Some families believed in animism which was the predominant belief of participant, recognized that they could be severely punished by the ancestor worship for "bad karma".

> *"My husband could not live with us because he cannot accept that his son has cerebral palsy (CP). He perceived that CP was his fault and felt very stigmatization. When my son turned 3 months, my husband moved to Malaysia for work, which is far from home. He said [my son's illness] was a consequence of his bad karma."*

> *"My mother said that our sister needs us and we have to care for her. . .it was not someone's fault. It might be the karma that she made in her previous life. . .if we leave her so. . .one day we might suffer like her. . ..so we have to help. . . and I believed I can take care of her as long as I could live."*

**2. Managing the in home-environment.** Thirty-two of 43 participants were frustrated when discussing their locations of residence that were far away from healthcare facilities. Participants talked about the effort to solve the problem step-by-step by seeking for the ways to care for the CSHCN, initially developing a plan of care at home and redesigning their roles and functions.

*Redesigning family functions.* The families of CSHCN perceived life with incurable illness as a confinement and they sought to release themselves from the limitations that imposed. One important idea, which emerged from the data, is that these families re-designed their functions and schedules under the supervision of the healthcare provider. The result was that they enhanced their ability to care for the CSHCN and shared the care thereby keeping the family together and improving resilience.

> *"there are six people in my family. . . grandparents, my husband, my son, my daughter, and me. . .I am a house worker and work with my husband in the field while grandparents cared for [my daughter] at home. Once I reach home I prepare food for the rest of the family and then I took her back [from the grandparents] . . .thiat's what we do every day. . . I believe it will get better".*

*Promoting care competency.* According to the 21st century care models, healthcare providers consider that family is the fundamental element for CSHCN, and that the home is an appropriate environment for care, therefore establishing or increasing care competency within the family is the first priority. Healthcare providers in Thailand design training courses for the family such as feeding, physiotherapy, and suctioning to meet the needs of the children. Most of the families reported that these programs increased their caring capacity, confidence, life satisfaction, and self-esteem.

> *"We live on the mountaintop which means we could not go to the hospital every day, so the nurse said that my house is the best place and I can take care for my child by working with the nurse or other healthcare providers. . .I feel better because I can save money and time. . .and my child still has good health."*

*"I try to follow the healthcare provider's advice. I asked them too many questions and I practiced procedures such as lung percussion, syringe ball suctioning, tube feeding, for my child. . .I think when I tried hard and I can do it."*

**3. Integrating care into a community health system.**   The primary care unit, SDHPH, plays a vital role in caring for CSHCN especially in rural areas, by focusing on promoting child health and providing continuing care under the Universal Coverage (UC) scheme. This valuable network maintains family satisfaction and community orientation of care.

*Developing community partnerships.* Caring for CSHCN in the home environment encourages nurse practitioners and public health officers to establish continuing care services such as home visits and mobile clinics to care for CSHCN at home. This means that care and support of CSHCN is the responsibility of the local government and municipal office. Community agencies are aware of, recognize and value the presence of CSHCN in their community. Some families choose to refer all the child's information to the community hospital or SDHPH to receive seamless care in their home environment.

*"Before I started working with the community, I had to take my child to the regional hospital which is far from my home when her (urine) bag was blocked. Once I called the nurse who works in our community and they came immediately to change the new bag for her . . .I felt like it was better for us. From that time, I contacted the community hospital and SDHPH to get my child into their services. This is a good thing in my life even if my child has CP because I have a good support."*

*Working with village healthcare volunteer (VHV).* VHV are lay people who are willing to provide health services to others in their community and to serve as connectors between the community and the healthcare providers in communities that lacked access to adequate care. VHV undergo basic health-training course at a SDHPH and on-the-job training working with healthcare providers. Through this process, VHVs are integrated into the health system, which also helps establish close cooperation between families, CSHCN and healthcare providers.

*"The VHVs are the voice for the people in the community. They work for SDHPH to care for the people in this area including my child, which makes me feel like there are more hands helping my child and our family."*

*"The close relationship with the community makes the VHVs a very important resource because they know our traditions well and understand the reality of the problems and can help with realistic solutions."*

**4. Maintaining family normalization.**   Caring for CSHCN is a difficult process and most of the participants reported that they live with difficulties, negative life events and challenges with respect to maintaining their family's function and normalcy. Some of the participants also addressed the power of faith and ancestor worship. In a mixture of Buddhist and animist beliefs, they believe that making merit to a person or to local spirits will bring them power to help them recover from the impact of discovering that their child has serious health condition.

*Valuing family members.* Family members were recognized and valued beyond their skills and ability to help manage illness. Most of the participants stated that they want to have every member of their family living together whether they remain healthy or develop an illness, and that they would assist each other without question.

*"I could tell you that every single person in my family helped me to care for my daughter. We live together, love each other, and pray for our members. Some people could not give an exact definition of the family but for me. . .family is everything. . .we share both good and bad situations."*

*"I believe that caring for the son with Down syndrome (DS) was not a problem for our family. Helping family members is a cultural tradition. We very much appreciate the way we live and that makes us share our time, place, and resources, and we all live together in the home we built."*

*Growing up with special conditions.* Cultural attitudes affected the way families care for the CSHCN in Thailand. Older caregivers such as grandparents delivered care to these CSHCH unconditionally while other member shared the responsibility with parents. It is a cultural tradition to maintain family integrity and resilience. The family felt that they were contributing good karma to their family and hoping that they would be able to change their circumstances in another life.

*"Fall fast then get up faster. . .We fall to learn how to rise and not fall again and we see every difficulty as the chance to test our resilience. . .our special ability."*

*"I believe in my lord Buddha, ancestor worship, and extraordinary power. They will help the person who tries to have good habits, attitudes, and manners. I told myself every day "do the good karma" for every situation with my heart and my spirit. That is the way I live for now."*

## Discussion

This study explored families' experience of caring for CSHCN in a rural area in Thailand. Adjusting family's life is the core phenomenon, in which engagement of personal beliefs, faith and religion, as well as support from family members motivated the family to provide care and to shape their future to live well. This model also discusses how to promote care competency for the family and healthcare service delivery at the community level, which is felt by families in this study as the way to provide the best possible care for CSHCN.

Adjusting family's life shows some similarities to the previous theoretical frameworks [3, 10, 20, 24–27] in the literature on caring experiences of families with children with complex conditions. Experience caring for the CSHCN is an unexpected life situation for the family. Increased assistance or supports at the very beginning are important because the incurable and long term conditions are cause of concern, stress, and burden. This study shows that families experience negative emotions based on the complexity of the disease, the lack of understanding of what was happening to their life [1, 2] and inequities in healthcare access in rural areas [28, 29]. Many families describe their emotional responses as angry or depressed, and some families end their marriage [6, 30–32] due to the caregiver burden, which is made worse if families are far from resources and support systems, or if they have financial problems [16, 33, 34]. However, not only were external factors reported but internal factors also affected their lives and experiences such as beliefs, culture and religion [35, 36]. In this study, most participants perceived that having a CSHCN resulted from ancestors' punishment or the consequences of bad karma from their previous life. This belief system may foster sensitivity and individualized care for CSHCN. Thus, when caring for a family with CSHCN, healthcare professionals should be attentive to and concerned with culture, beliefs, traditions and practices, which can help them gain an understanding and tailor early interventions, and prepare for transitions to adulthood for CSHCN within their contexts and environments [5, 10, 37].

The second important concept was that of managing to care for CSHCN at home. All family members are of key importance in helping to overcome challenges and maintain normalcy in their life in the context of living in a rural area which significantly determines how a family lives their life [38, 39]. In previous studies, when family members are ill or hospitalized, the family primarily receives support from mothers [10, 40], and some of them have articulated feelings of anxiety or worry when healthcare professionals teach them procedures such as wound dressing or feeding [41–43]. In contrast, in the current study, the caregiver role was distributed among other members especially grandparents, and siblings. They initially sought assistance from nurses, public health officer and VHV in order to learn to provide care in ways that were appropriate to their traditions, beliefs and lifestyle, even when some of them were illiterate [24, 44, 45]. This finding illustrates that the family would prefer to have agency and to control their own schedule at home [24, 46] rather than to depend on healthcare providers [47–49]. Studies on CSHCN including conditions such as asthma, autism spectrum disorder, and diabetes have found that using a home-based intervention, not only improved child health and quality of life, but also relieved psychological distress within the family [50].

Some Southeast Asian countries, including Thailand, have launched universal coverage policies for a decade. Nevertheless, accessibility and quality of care have been identified as challenges in prior studies [28, 29, 49, 50]. Due to geographical challenges, some health services and facilities were not available to the families in our study. Thus, integrating care for CSHCN into the community health system was the third construct discussed by families in this study. Our study found that to bridge the equity gap in access to care, healthcare providers have to be available across their catchment area to provide high quality and satisfactory care to the family of CSHCN in person or using digital technologies [50, 51]. However, only privileged people are likely to use smart devices, which means that access may be limited for the majority of people who have low literacy and financial limitations. Despite the lack of technology, one surprising finding in the study was the importance of VHV involvement as a facilitator, supporter, and coordinator of the medical system between the families and communities and the healthcare system [52]. The families in our study reported positive experiences when worked closely with VHV who have engaged in CHSCH care. VHVs may be able to enlist cooperation and achieve mutual understanding between the family and the healthcare system and might also have a sense of community responsibility and provide a point of entry for healthcare professionals [37, 52, 53]. VHV involvement not only benefits the family but also increases the capacity of the community health system to provide care for both CSHCN and their families across the complex needs or diseases and is identified as a keystone of this system [26, 47, 51].

The final concept shared was the importance of maintaining family normalization. This study reinforces previous insights on the role of religion, beliefs, and traditional practices in contributing to disease tolerance and added new understanding about caring for CSHCN in rural areas [39, 54–57]. Most of the families in our study articulated religious beliefs, including mixtures of Buddhism, ancestor worship, and animism [38, 58, 59]. Buddhism dictates that present actions will affect the future, meaning that doing good will having good outcomes. Ancestor worship is founded on the belief that deceased family members are guardians, which protect the whole family from illness and death. The ancestor spirits are still present to observe and one should do good deeds and think good thoughts about ones family [59, 60]. Believing in this protection, as well as understanding all family members past and present as being an active part of the life of the family are important factors in maintaining an overall positive outlook, family integrity, and support for the CSHCN.

## Strengths and limitations

All participants in this study were people who have lived experience caring for CSHCN in a rural area in Thailand. The participants showed how they were working together in caring for CSHCN in rural areas. They mutually managed care situations and maintained their function and normalcy. Above this, as healthcare professionals, we need to take into account that caring for CSHCN is a dynamic and systemic process that requires supports and resources to allow families to adjust their life throughout the life course of the CSHCN.

The limited interview duration may have resulted in an inadequate reflection of a whole lifetime of caring for the CHSCN, some aspects might have been overlooked, and the sample size was small. In addition, quantitative investigations should be conducted in the future to further elucidate the factors constructed in this model.

## Conclusion

This study is a comprehensive explanation of the experiences of families caring for CSHCN in a rural area in Thailand, including an examination of family members' roles and responsibilities. Adjusting family's life is the emergent theory that highlights how families manage their lives and functions, how they seek support from each other and from the healthcare system, and the strategies they use to maintain their everyday lives. This theory offers guidance that can be used to develop training programs for healthcare providers, improving patient and family-centered care in primary care and other ambulatory settings, as well as potentially being integrated into international classification framework (ICF) to promote a holistic approach in transition planning, scale up successful interventions, and fill crucial gaps in caring for CSHCN in rural areas. Additionally, this theory can also contribute to policy or changes to service provision, and can alter thinking in order to initiate change in this substantive area of inquiry.

## Supporting information

**S1 File. Semi-structured interview guide in English.**
(DOCX)

## Acknowledgments

We are also grateful to all the family members who participated in this study and the health care professionals who work in rural area for their assistance in getting access to and recruiting participants. Thank you school of Nursing, Mae Fah Luang University for excellent supporting.

## Author Contributions

**Conceptualization:** Katemanee Moonpanane, Eva Purkey.

**Data curation:** Salisa Kodyee.

**Formal analysis:** Katemanee Moonpanane.

**Funding acquisition:** Katemanee Moonpanane.

**Investigation:** Katemanee Moonpanane, Salisa Kodyee.

**Methodology:** Katemanee Moonpanane, Salisa Kodyee.

**Project administration:** Katemanee Moonpanane, Salisa Kodyee.

**Resources:** Chomnard Potjanamart.

**Supervision:** Chomnard Potjanamart, Eva Purkey.

**Validation:** Katemanee Moonpanane, Salisa Kodyee, Eva Purkey.

**Visualization:** Chomnard Potjanamart.

**Writing – original draft:** Katemanee Moonpanane.

**Writing – review & editing:** Katemanee Moonpanane, Chomnard Potjanamart, Eva Purkey.

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
