## [Decision Letter · Decision Letter 0]

14 Jun 2021

PONE-D-21-08488

Adjusting the family’s life: A grounded theory of caring for children with special healthcare needs in rural areas, Thailand.

PLOS ONE

Dear Dr. katemanee moonpanane,

Thank you for submitting your manuscript to PLOS ONE. After careful consideration, we feel that it has merit but does not fully meet PLOS ONE’s publication criteria as it currently stands. Therefore, we invite you to submit a revised version of the manuscript that addresses the points raised during the review process.

We look forward to receiving your revised manuscript.

Kind regards,

Sharon Mary Brownie

Academic Editor

PLOS ONE

Journal Requirements:

3. Please include your tables as part of your main manuscript and remove the individual files. Please note that supplementary tables should  be uploaded as separate "supporting information" files.

Reviewers' comments:

Reviewer's Responses to Questions

**Comments to the Author**

1. Is the manuscript technically sound, and do the data support the conclusions?

Reviewer #1: Yes

Reviewer #2: Partly

2. Has the statistical analysis been performed appropriately and rigorously? 

Reviewer #1: Yes

Reviewer #2: N/A

3. Have the authors made all data underlying the findings in their manuscript fully available?

Reviewer #1: Yes

Reviewer #2: No

4. Is the manuscript presented in an intelligible fashion and written in standard English?

Reviewer #1: Yes

Reviewer #2: Yes

5. Review Comments to the Author

Reviewer #1: Recommendations:

In Background part

Infectious disease instead of infection disease

In Methods

Forty-three family members of CHSCN participated in in-depth interviews. How many families the proposal studied, meaning those 43 members representing how many families??? Kindly add to methods.

The main source of the data that support this research are the interviews carried out from almost all different family members. However, reviewer recommend to highlight and analyze separately care giving mothers, who experience the work of caring for their children on a daily basis , in order to ensure the trustworthiness of the results.

Factors affecting the implementation of digital health programs should be discussed in relation to technological, human, and systems factors connecting patients and their families to the needed care from expert health care providers in particular to families in rural areas.

Reviewer highly recommend authors to highlights home (in persons appointments) or Telehealth Interventions to Teach and Support Parents and Family members especially for children with Autism Spectrum Disorders to reduce their problem behavior.

Reviewer #2: Overall this is a good study and needed for a developing country like Thailand. The English language used is OK. However, I have several issues which are unclear to me:

1. Interviews were conducted with these 43 families. How many family members were being interviewed in each family? Or the 43 participants were representatives from 43 families (i.e. 1 participant per family)?

2. I am also not clear whether you conducted the interview for one family at a time or you conducted several focus group discussions to extract the information?

3. Participants were from a rural area - does this mean, all participants were from one village or area? I don't think so because it can't be 43 families in one village area had children with special needs. Maybe you need to explain the demographic of this 'a rural area' where you collected your data so that it would be easier to understand the situation.

4. Strategies used to strengthen the study rigor should be explained in the methodology of how you processed the interview data.

5. Line 181 mentioned about 'most of the participants'... - please specify how many?

6. Line 182 - 'They talked about...' - It was not clear 'they' refers to who?

7. Line 213 - The primary acre unit..' - what is this? Typo error? I figured out that could be 'care'. What does SDHPH stands for? Write in full when you mentioned this for the first time in the text.

8. Line 287 - The statement 'previous theoretical frameworks...' - please indicate the references here.

9. Line 318 - Again, no references given to the statement '...in prior studies'. Please insert the references here - which studies were referred to.

10. This emergent theory 'Adjusting the family's life' can also be discussed within the framework of the International Classification of Functioning (ICF) third party in which disability of a family member affect the others in the family. You may want to discuss about this third party ICF in your discussion and conclusion.

6. PLOS authors have the option to publish the peer review history of their article (what does this mean?). If published, this will include your full peer review and any attached files.

Reviewer #1: **Yes: **Prof. Nagwa A Meguid

Reviewer #2: No

---

## [Author Response · Author response to Decision Letter 0]

22 Sep 2021

Response to Reviewer’s Comments

Dear Editor,

Thank you very much for the opportunity to submit the revised manuscript title “Adjusting the family’s life: A grounded theory of caring for children with special health care needs in rural, Thailand” that was submitted to PLOS One. Coauthors and I very much appreciated the encouraging and critical comments on this manuscript by the reviewers. The comments have been very thorough and helpful in improving the manuscript. We strongly believe that the comments and suggestions have increased the value of revised manuscript by many folds. 

Here is point-by-point response to the reviewers’ comments and concerns.

Best Regards,

Katemanee Moonpanane

September 21, 2021

Reviewer #1: Recommendations:

1. In Background part

Infectious disease instead of infection disease

Response: Sorry for using the wrong word. It has been replaced with “infectious disease” as recommended please see line 32. Thank you so much.

2. In Methods

Forty-three family members of CHSCN participated in in-depth interviews. How many families the proposal studied, meaning those 43 members representing how many families??? Kindly add to methods.

Response: Thank you very much for the comment, which addresses the point that was not covered in the method, study population section. The manuscript has been revised as suggested; please see (lines 95-98).

- The main source of the data that support this research are the interviews carried out from almost all different family members. However, reviewer recommend to highlight and analyze separately care giving mothers, who experience the work of caring for their children on a daily basis, in order to ensure the trustworthiness of the results.

Response: Thank you for valuable suggestion and we agree on this point. Mothers play an important role for the CSHCN on daily basis, but in Thailand, parenting roles attribute to other family members especially grandmothers and siblings and these caring pattern originated the research project. Then the variety of participants were selected (15 mother, 12- grandmothers and 8-female siblings). We believed the finding of the study present the real caring experiences and processes of the family members who live with CSHCN in rural Thailand. However, we have added information in the Rigor section (lines 149-166) to robust and ensure the trustworthiness. I apologize if my response may not directly address your comment. 

- Factors affecting the implementation of digital health programs should be discussed in relation to technological, human, and systems factors connecting patients and their families to the needed care from expert health care providers in particular to families in rural areas.

Response: Thank you for the helpful comment, there are some groups of people in rural Thailand who use the digital technologies more than the majority group who have the limited financial resources. However, we made the relevant issue of digital technologies prominent in the revised version. Please see lines 375-383.

- Reviewer highly recommend authors to highlights home (in persons appointments) or Telehealth Interventions to Teach and Support Parents and Family members especially for children with Autism Spectrum Disorders to reduce their problem behavior.

Response: Thank you for excellent comment. We have added the relevant issue related to home intervention for CSHCN included ASD in recommendation section. Please see lines 364-369.

Reviewer #2: Overall this is a good study and needed for a developing country like Thailand. The English language used is OK. However, I have several issues, which are unclear to me:

1. Interviews were conducted with these 43 families. How many family members were being interviewed in each family? Or the 43 participants were representatives from 43 families (i.e. 1 participant per family)?

Response: Thank you for the opportunity to provide further clarification. Additional information has been added; lines 111-117 and 169-175.

2. I am also not clear whether you conducted the interview for one family at a time or you conducted several focus group discussions to extract the information?

Response: Thank you for raising an important point. Reviewer’s suggestion is incorporated in the revised version of manuscript; please see lines 111-117.

3. Participants were from a rural area - does this mean, all participants were from one village or area? I don't think so because it can't be 43 families in one village area had children with special needs. Maybe you need to explain the demographic of this 'a rural area' where you collected your data so that it would be easier to understand the situation.

Response: Thank you for your suggestion; we have added information in the Setting section. Please see lines 75-81.

4. Strategies used to strengthen the study rigor should be explained in the methodology of how you processed the interview data.

Response: Thank you for your suggestion, we look so long time to look over the whole information from setting to rigor and repeat the step of analysis. we also used a long time to look inside the information in this section and putting additional information in Data collection (lines 125-131) and Rigor section (lines 149-166).

5. Line 181 mentioned about 'most of the participants'... - please specify how many?

Response: Thank you for the comment. The sentence is reconstructed to make it more clear for the reader, please see line 222.

6. Line 182 - 'They talked about...' - It was not clear 'they' refers to who?

Response: Thank you for the comment. It has been replaced with the “the participants” please see line 223.

7. Line 213 - The primary acre unit..' - what is this? Typo error? I figured out that could be 'care'. What does SDHPH stands for? Write in full when you mentioned this for the first time in the text.

Response: Sorry for the wrong spelling. It was replaced with “primary care unit” (line 256) and the full form of SDHPH is stand for Sub-district health promotion hospital (SDHPH) and it was mention firstly in line 87.

8. Line 287 - The statement 'previous theoretical frameworks...' - please indicate the references here.

Response: Thank you for your suggestion, the relevant literature had added, please see in lines 332-333.

9. Line 318 - Again, no references given to the statement '...in prior studies'. Please insert the references here - which studies were referred to.

Response: Thank you for your suggestion, the relevant literature had added, please see in line 372.

10. This emergent theory 'Adjusting the family's life' can also be discussed within the framework of the International Classification of Functioning (ICF) third party in which disability of a family member affect the others in the family. You may want to discuss about this third party ICF in your discussion and conclusion.

Response: Thank you for your suggestion, ICF model provide a guideline for child functioning with an emphasis on child and family participation but in Thailand ICF model is now only using for the child with disability and most of healthcare professional were not familiar with ICF model. However, we added the relevant information in Discussion (lines 348-352, 385-388) and in Conclusion (lines 426-431) as the recommendation. 

Once again, I wish to thank you the reviewers for taking time to provide the excellent comments and suggestions. We hope that our revised version manuscript meet your expectations. We hope that once the paper has been published, the content, including the suggestion and recommendation will reach those who need it. 

Best Regards,

Katemanee Moonpanane

---

## [Decision Letter · Decision Letter 1]

4 Oct 2021

Adjusting the family’s life: A grounded theory of caring for children with special healthcare needs in rural areas, Thailand.

PONE-D-21-08488R1

Dear Dr. katemanee moonpanane,

We’re pleased to inform you that your manuscript has been judged scientifically suitable for publication and will be formally accepted for publication once it meets all outstanding technical requirements.

Kind regards,

Sharon Mary Brownie

Academic Editor

PLOS ONE

 Editor Comments 

Reviewer recommendations have been satisfactorily addressed.

Reviewers' comments:

Reviewer's Responses to Questions

**Comments to the Author**

1. If the authors have adequately addressed your comments raised in a previous round of review and you feel that this manuscript is now acceptable for publication, you may indicate that here to bypass the “Comments to the Author” section, enter your conflict of interest statement in the “Confidential to Editor” section, and submit your "Accept" recommendation.

Reviewer #1: All comments have been addressed

2. Is the manuscript technically sound, and do the data support the conclusions?

Reviewer #1: Yes

3. Has the statistical analysis been performed appropriately and rigorously? 

Reviewer #1: Yes

4. Have the authors made all data underlying the findings in their manuscript fully available?

Reviewer #1: Yes

5. Is the manuscript presented in an intelligible fashion and written in standard English?

Reviewer #1: Yes

6. Review Comments to the Author

Reviewer #1: Authors have adequately addressed your comments and manuscript technically sound. Manuscript is presented in an intelligible fashion and written in standard English

7. PLOS authors have the option to publish the peer review history of their article (what does this mean?). If published, this will include your full peer review and any attached files.

Reviewer #1: **Yes: **Dr Nagwa Abdel Meguid, Prof. of Human Genetics at National Research Centre, Egypt

---

## [Editor Report · Acceptance letter]

15 Oct 2021

PONE-D-21-08488R1 

Adjusting the family’s life: A grounded theory of caring for children with special healthcare needs in rural areas, Thailand. 

Dear Dr. Moonpanane:

I'm pleased to inform you that your manuscript has been deemed suitable for publication in PLOS ONE. Congratulations! Your manuscript is now with our production department. 

Kind regards, 

on behalf of

Professor Sharon Mary Brownie 

Academic Editor

PLOS ONE